# Dual properties of a hydrogen oxidation Ni-catalyst entrapped within a polymer promote self-defense against oxygen

Alaa A. Oughli[1], Adrian Ruff[2], Nilusha Priyadarshani Boralugodage[3], Patricia Rodríguez-Maciá[1], Nicolas Plumeré [4], Wolfgang Lubitz[1], Wendy J. Shaw[3], Wolfgang Schuhmann [2] & Olaf Rüdiger [1]

The $Ni(P_2N_2)_2$ catalysts are among the most efficient non-noble-metal based molecular catalysts for $H_2$ cycling. However, these catalysts are $O_2$ sensitive and lack long term stability under operating conditions. Here, we show that in a redox silent polymer matrix the catalyst is dispersed into two functionally different reaction layers. Close to the electrode surface is the "active" layer where the catalyst oxidizes $H_2$ and exchanges electrons with the electrode generating a current. At the outer film boundary, insulation of the catalyst from the electrode forms a "protection" layer in which $H_2$ is used by the catalyst to convert $O_2$ to $H_2O$, thereby providing the "active" layer with a barrier against $O_2$. This simple but efficient polymer-based electrode design solves one of the biggest limitations of these otherwise very efficient catalysts enhancing its stability for catalytic $H_2$ oxidation as well as $O_2$ tolerance.

---

[1] Max-Planck-Institut for Chemical Energy Conversion, Stiftstrasse 34–36, 45470 Mülheim an der Ruhr, Germany. [2] Department Analytical Chemistry, Center for Electrochemical Sciences (CES), Ruhr-Universität Bochum, Universitätsstrasse 150, 44780 Bochum, Germany. [3] Pacific Northwest National Laboratory, 902 Battelle Blvd, Richland, WA 99352, USA. [4] Center for Electrochemical Sciences—Molecular Nanostructures, Ruhr-Universität Bochum, Universitätsstrasse 150, 44780 Bochum, Germany. Correspondence and requests for materials should be addressed to W.J.S. (email: Wendy.Shaw@pnnl.gov) or to W.S. (email: wolfgang.schuhmann@rub.de) or to O.Rüd. (email: olaf.ruediger@cec.mpg.de)

A major challenge of humankind is a future sustainable energy economy. Hydrogen has been proposed as an ideal target to store energy from renewable sources, e.g., solar-driven water splitting[1–4]. A hydrogen-powered fuel cell is then able to recover a major part of the energy in high yields on demand. An unsolved challenge in this endeavor is to design active, efficient and stable catalysts based on earth-abundant metals[4].

Nature uses the highly active and efficient hydrogenase enzymes for hydrogen cycling in living systems. The enzymes are capable of achieving low overpotentials and high turnover frequencies in both hydrogen oxidation and proton reduction, bearing only the abundant metals Fe and/or Ni in their active sites[5]. Chemists have been inspired by these enzymes to design inexpensive molecular complexes capable of efficiently producing or oxidizing $H_2$. Particularly noteworthy examples are the DuBois catalysts[6,7], which are Ni-(bis)diphosphine based complexes equipped with a pendant amine that acts as a Lewis base, accepting the proton during $H_2$ splitting, in a manner analogous to that proposed to occur at the active site of [FeFe] hydrogenases[8,9].

Extension of the proton channel with a carboxylic acid moiety between the metal center and the solvent by attachment of an amino acid to the pendant amine, allows the DuBois catalyst to operate at very low overpotentials at low pH and at room temperature in aqueous systems[10–12], or even reversibly[13,14]. Moreover, these catalysts show tolerance towards CO, a common contaminant of $H_2$ feedstocks and a strong inhibitor of hydrogenases and of platinum used for catalysis[5,15,16].

On the other hand, the DuBois catalyst undergoes rapid degradation under conditions relevant for technological applications, i.e., when immobilized on an electrode[15,17,18] or on photoactive materials[19]. We demonstrated in a previous study that the electrocatalytic $H_2$ oxidation activity of the DuBois catalyst CyGly (for Structure see Fig. 1) was lost irreversibly after 10 min when 2% $O_2$ was added to the $H_2$ gas feed[15]. Similarly, other studies with DuBois complexes in solution revealed that the $H_2$ production activity was completely suppressed when $O_2$ was present in the electrochemical cell[16,20]. Crystallographic and nuclear magnetic resonance (NMR) spectroscopic studies showed that the Ni-(bis)diphosphine complexes in low oxidation states react with $O_2$ to oxidize the phosphine ligands, inactivating the

complexes irreversibly[21]. Hence, under electrocatalytic conditions implying fast collection of electrons by the electrode, which is desired for current generation in fuel cells, exacerbates the $O_2$ sensitivity of the Ni-catalyst. In contrast, in solution and in the presence of $H_2$, the Ni-complex accumulates in a highly reduced doubly protonated $Ni^0$ state, which can catalytically reduce $O_2$ to water, slowing the formation of the oxidized inactive complex (Fig. 1)[21].

In the present study, we take advantage of the dual catalytic nature of this complex: the electrocatalytic oxidation of $H_2$ and the catalytic reduction of $O_2$. We use here the properties of a redox-silent hydrophobic polymer to support the catalyst and obtain a polymer/catalyst film which creates two discrete reaction layers serving separate roles: a phase close to the electrode surface where electrons can directly tunnel between catalyst and electrode. This layer is responsible for electrocatalytic $H_2$ oxidation. Near the interface between the polymer and the electrolyte, a second region which is electrically disconnected from the electrode surface, prevents anodic reoxidation of the catalyst, allowing the doubly protonated $Ni^0$ complex to reduce incoming $O_2$ to water. This results in a simple and efficient self-protecting, catalytically active matrix that serves to extend the limited lifetime of the Ni-catalyst when immobilized on an electrode surface.

## Results

**Reactivity of the CyGly complex on electrode surfaces with $O_2$.** The catalyst was dispersed into a poly(glycidyl methacrylate-co-butyl acrylate-co-poly(ethylene glycol) methacrylate) (denoted as P(GMA-BA-PEGMA) in the following) polymer (See Methods section)). This polymer was chosen because of its rather hydrophobic nature that allows for the formation of stable films in aqueous electrolytes. Moreover, the polymer backbone consists of chemically inert and non-coordinating monomers. Thus, unintended interactions between the Ni-center and the polymer backbone that may affect the catalytic properties of the catalyst are prevented. Film formation was carried out by drop-casting of a homogenous solution of the polymer and the complex in acetonitrile on a glassy carbon electrode followed by drying under anaerobic conditions. Consecutive cyclic voltammograms of the modified electrodes at pH 3.0 revealed the appearance of a reversible signal with a mid-point potential of −12 mV vs. SHE (Fig. 2). This wave was previously assigned to two overlapping one electron processes, corresponding to the reduction of $Ni^{2+}$ to $Ni^0$ [10,15]. The intensity of the signal increases during the first 30 min as a result of the initial solvation of the polymer and/or penetration of counter ions into the film (see Supplementary Fig. 1). The peak intensity is proportional to the square root of the scan rate, indicating the diffusional nature of the electrochemical process. This is indicative of some mobility of the Ni-complex, and/or counter ion transport inside the polymer film (See Supplementary Fig. 2). When $H_2$ is flushed through the electrochemical cell, a catalytic current appears starting at the redox potential of the complex (Fig. 2b). The catalytic current increases with the applied potential, indicating slow counter ion movement within the film or a distribution of the tunneling distance between the CyGly molecules and the electrode as a result of the non-conductive nature of the polymer. The latter is similar to what has been reported for complexes in solution with bulky dipeptides in the outer coordination sphere[22].

Addition of $O_2$ to the gas mixture during $H_2$ oxidation using a film of CyGly complex dispersed in the polymer, does not affect the catalytic current (Fig. 2c, black trace). This is in stark contrast to measurements with a monolayer of the immobilized catalyst, where the CyGly complex loses 75% of its catalytic $H_2$ oxidation

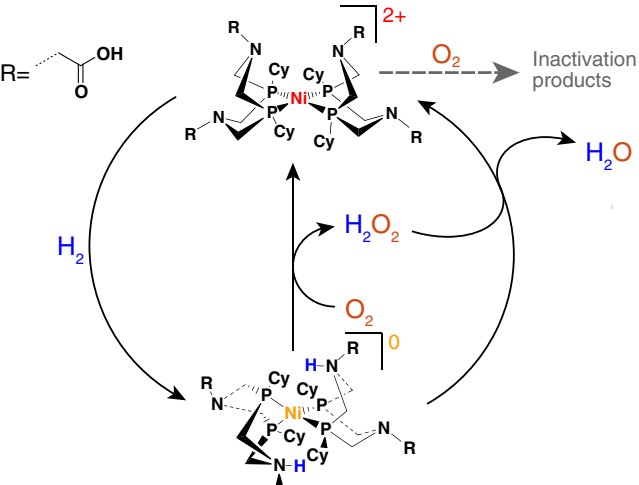

**Fig. 1** Reactions of the CyGly complex with $H_2/O_2$ in solution. The $Ni^{2+}$ complex, $[Ni^{II}(P^{Cy}_2N^{Gly}_2)_2]^{2+}$, undergoes slow and irreversible inactivation by $O_2$ while the $H_2$ reduced $Ni^0$ complex, $[Ni^0(P^{Cy}_2N^{Gly}_2)_2]$, is able to catalytically reduce $O_2$ to $H_2O$

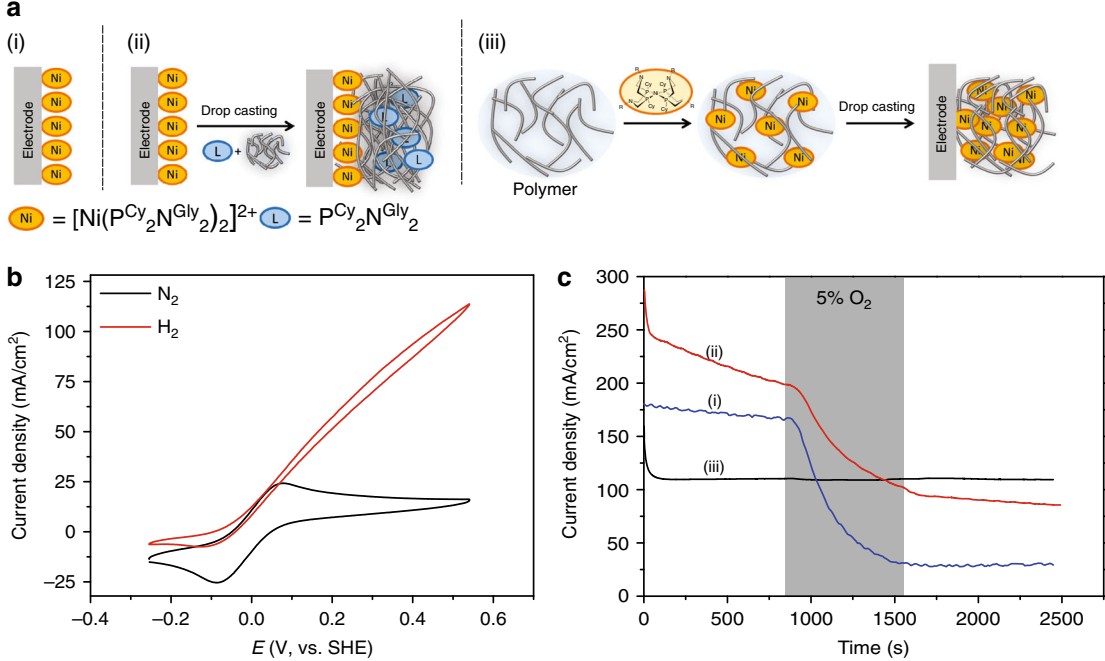

**Fig. 2** Catalytic properties and tolerance of CyGly modified electrodes towards oxygen. **a** Electrode preparation schemes: (i) A monolayer of CyGly complex; (ii) a monolayer of CyGly complex coated with the pristine polymer and the $P^{Cy}_2N^{Gly}_2$ ligand, and (iii) CyGly/polymer film on glassy carbon electrodes. **b** Cyclic voltammograms of the polymer/catalyst film (iii) under $N_2$ (black trace) and under $H_2$ (red trace). Conditions: 20 mV s$^{-1}$, pH = 3, 0.1 M MES/0.1 M HEPES + HClO$_4$, 25 °C, 2000 rpm. **c** Chronoamperometry experiments under 90% $H_2$/10% $N_2$ with the addition of 5% $O_2$ (90% $H_2$, 5% $N_2$) over 600 s (shaded area) for all three electrode preparations. Conditions: + 541 mV vs. SHE, 25 °C, 2000 rpm, pH = 3, 0.1 M MES/0.1 M HEPES + HClO$_4$. The initial current drop corresponds to slow equilibration processes inside the polymer (establishment of $H_2$ and Ni$^{2+}$ gradients)

activity irreversibly over 10 min as 5% $O_2$ is added to the gas feed (Fig. 2c, blue trace). On the other hand, when the catalyst in a DET configuration is exposed to $O_2$ in the absence of $H_2$, most of the catalytic activity is recovered when the gas is switched back to $H_2$ (See Supplementary Fig. 3b, blue trace). This suggests a higher $O_2$ sensitivity of the complex during electrocatalytic $H_2$ oxidation. When the potential on the electrode was kept low enough to maintain the Ni-center in the reduced Ni$^0$ state during $O_2$ exposure, the catalytic current was completely recovered (see Supplementary Fig. 3b, red trace). In line with this result, electrodes prepared outside the glovebox did not show any catalytic activity (data not shown) suggesting that long exposure of the Ni-complex in the Ni$^{2+}$ oxidation state to $O_2$ without any applied potential on the electrode or in the absence of $H_2$ damages the catalytic properties of the complex.

To evaluate if the observed protection is only based on a simple physical blocking of $O_2$ by the polymer, cyclic voltammetry experiments with polymer-coated glassy carbon electrodes in the presence of $O_2$ were conducted. Indeed, direct $O_2$ reduction at glassy carbon electrodes is significantly hampered by the presence of the polymer film on the electrode surface (See Supplementary Fig. 4). However, when a monolayer of the catalyst on the electrode surface was further coated with a mixture of polymer and ligand (of similar thickness, with two equivalents of CyGly ligand but without Ni), the catalyst exhibits similar sensitivity to oxygen as the monolayer without a polymer top layer (Fig. 2c, red trace). This excludes on one hand the possibility that a physical barrier to oxygen is the sole explanation for the decreased oxygen sensitivity in catalyst-polymer films. On the other hand, since the CyGly ligand is also $O_2$ sensitive, this experiment further demonstrates that the simple stoichiometric reaction of $O_2$ with the catalyst during its oxidative degradation does not significantly participate in the protection mechanism. An active Ni-complex dispersed in the polymer film is required for effective protection.

**The CyGly complex as an $O_2$ reducing catalyst**. The reactivity of the CyGly complex with $O_2$ in solution was analyzed by means of $^2H$ NMR spectroscopy. $D_2$ was used as a reductant and the formation of $D_2O$ upon $O_2$ addition to the gas feed was quantified. The TOF for $O_2$ reduction to $D_2O$ was measured to be $(20 \pm 5)$ h$^{-1}$ (see Supplementary Fig. 5). Formation of $D_2O$ under such conditions supports the mechanism proposed in Fig. 1 and is consistent with results described previously by Yang et al[21]. for a series of similar complexes containing a pendant base on the phosphine ligand (see Supplementary Note 1).

On the other hand, if the NMR tube contains mixtures of $H_2$ and $O_2$ in order to reproduce the conditions in the electrochemical cell, the $^{31}P$ NMR spectrum of the catalyst in solution shows almost complete loss of the CyGly resonances after 3 h along with new resonances of its degradation products appearing (See Supplementary Figs 11 and 12).

To reduce $O_2$ to water, two molecules of the Ni$^0$ complex are needed to provide the required 4 electrons and protons (Fig. 1). This is possible with the freely diffusing catalyst in solution, but using a monolayer of catalyst immobilized on the electrode, the catalyst is cycling through different redox states and geometries under turnover conditions[23]. As a result, the immobilized doubly protonated Ni$^0$ complex does not accumulate in the required concentrations to effectively reduce $O_2$ to water, because of fast electron transfer to the electrode. Consequently, catalyst degradation and the observed drop in catalytic current results (Fig. 2c, red and blue traces).

**A polymer matrix to separate reaction layers**. The stability of the $H_2$ oxidation currents under $O_2$ observed in Fig. 2c for the CyGly/polymer films can be explained if we consider a layered structure for the CyGly complex/polymer film. The insulating polymer acts as a supporting matrix to immobilize the complex

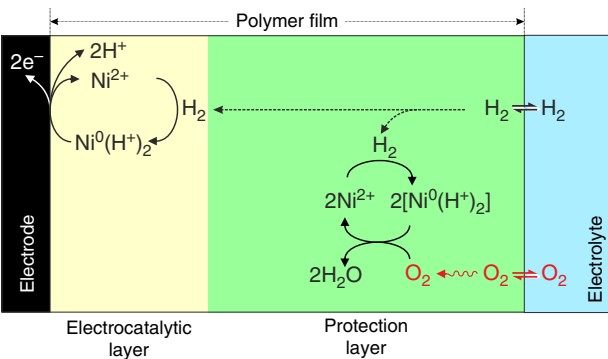

**Fig. 3** Proposed reactions inside the polymer/catalyst film. Schematic representation of the polymer film separating the immobilized CyGly complex into two functional layers where different reactions occur; colors are used to identify the different polymer layers and the boundary between polymer and electrolyte. In the electrocatalytic layer (yellow), $H_2$ is oxidized to protons and electrons via the Ni-complex. Within the protection layer (green) the doubly reduced $Ni^0$ complex catalyzes the reduction of incoming $O_2$ to $H_2O$ by using the electrons supplied from $H_2$

on the electrode surface, but also prevents electrical contact between the complexes in the remote regions of the film and the electrode. The formation of a sufficiently thick polymer film generates two distinct regions; with electrocatalytic $H_2$ oxidation in the inner layer near the electrode surface and $O_2$ reduction in the outer layer as shown in Fig. 3.

Under $H_2$ in the absence of $O_2$, the complex in the protection layer (Fig. 3, green region) is reduced to the protonated $Ni^0$ state, but since it is electrically isolated from the electrode, electrons will not be shuttled to the electrode. However, when $O_2$ reaches the film/electrolyte interface, it will encounter the reduced Ni-complexes, which will then reduce incoming $O_2$ to water, thus protecting the electrocatalytic layer (Fig. 3, yellow region) from $O_2$ damage.

**Validation of the protection mechanism**. The mechanism of $O_2$ reduction to water requires two catalyst molecules per $O_2$, which has implications for the film preparation. If films with low catalyst loading are used, oxidative damage of the catalyst is observed (See Supplementary Fig. 6). This is thought to be due to slow diffusion of the catalyst molecules within the film, thus allowing $O_2$ to pass the protection layer and/or the accumulation of partially reduced oxygen species.

Conductive multi-walled carbon nanotubes (MWCNT) were used to extend the electrocatalytic region in the polymer film, connecting the electrode to catalyst molecules in the film further away from the electrode, i.e., in the previously defined protection layer. The resulting electrode showed higher $H_2$ oxidation currents, but as expected, reduced the protection layer by increasing the number of catalyst molecules capable of $H_2$ electrocatalysis rather than $O_2$ reduction. Consequently, the result is a catalytic current decay when $O_2$ is added to the gas feed (see Supplementary Fig. 7). Moreover, the thickness of the film is crucial for effective protection. While thinner films produced higher catalytic currents for $H_2$ oxidation (substrate diffusion is not limiting), these films do not show protection against $O_2$ and the $H_2$ oxidation current decayed as soon as $O_2$ was added to the gas feed (Fig. 4a). Similar $O_2$ sensitivity as a function of film thickness was observed for an air sensitive enzyme, i.e., a NiFeSe hydrogenase, that was immobilized in an $O_2$ quenching redox polymer[24].

To further investigate the catalytic nature of the $O_2$ reduction reaction by the catalyst, we tested the stability of the catalytic

current when the film was exposed to $O_2$ in the absence of $H_2$. This gas is required to regenerate the reduced Ni-complex in the protection layer once it is oxidized by an $O_2$ molecule. Supplementary Fig. 8 shows that when the catalyst-polymer film is exposed to 5% $O_2$ in the absence of $H_2$ for 5 h, 35% of the catalytic activity is lost, while when $H_2$ is present, 100% of the initial current is maintained for the same $O_2$ exposure. Interestingly, while measuring the control experiment under $N_2$ (See Supplementary Fig. 8, blue trace), without $O_2$ to corroborate that $N_2$ is not damaging the complex, we noticed that the oxidation current did not reach 0 A. The most likely explanation for this observation is that as we start the experiment under $H_2$ to record the initial activity of the film, the catalyst in the polymer layer is completely reduced by $H_2$. Since diffusion inside the polymer is extremely slow, its oxidation by the electrode surface is also very slow. Integration of the charge passed gives a value of 31 µC, which would correspond to 0.2 µmol of Ni-complex, 20% of the catalyst present in the film. This explains why only 35% of the catalytic activity is lost under a $N_2/O_2$ mixture. Under such conditions, the Ni-complex in the polymer film is still reduced, and therefore protecting the complex in the inner layer, but in the absence of $H_2$ to regenerate the reduced complex, the inactive front advances more rapidly.

**The polymer enhances catalyst stability**. The catalytic current was remarkably stable over time. After 18 days of continuous measurement in a chronoamperometry experiment under turnover conditions in the absence of $O_2$, the electrode still maintained 75% of the initial current (See Supplementary Fig. 9). This stability contrasts with what was reported previously for a monolayer of the same catalyst covalently attached to the electrode, where the activity was completely lost after 3 days[15]. The increased stability could be a result of the hydrophobic nature of the polymer film, which may stabilize the integrity of the complex during turnover. Moreover, within the polymer film, the Ni-catalyst can freely diffuse without any structural and/or conformational constraints, as it is the case for the surface confined Ni-catalyst (See Supplementary Note 2).

The $H_2$ oxidation current remains completely unaffected by the presence of $O_2$ in the gas feed for the first 7 hours (Fig. 4b). After this time, the current starts to decay slowly, maintaining 75% of the initial current after 24 h of continuous $O_2$ exposure. As stated before, we cannot eliminate the undesired slow oxidation of the phosphines[21] forming an inactive product and therefore eliminating the protective capacity of the protection layer over time. After 7 h the catalyst in the protection layer is most likely starting to get oxidized, which results in an advancement of the inactive front reaching the catalyst molecules in the electrocatalytic layer. At this point, catalyst molecules participating in the electrocatalytic $H_2$ oxidation begin to decompose due to incoming $O_2$, resulting in the slow decay of the catalytic current.

## Discussion

DuBois catalysts with modifications in the outer coordination sphere have already been proposed as good candidates for use in fuel cells[10,14,23,25,26]. They operate at very low overpotential and unlike noble metals or hydrogenases they are insensitive to CO,[16,17] which allows for the use of inexpensive low purity $H_2$ as fuel. The remaining limitations to be solved are the lack of stability and the oxygen sensitivity[16]. We demonstrate that the use of a redox silent polymer acting as a support for the fabrication of $H_2$ oxidation may overcome these limitations. The polymer matrix provided several important advantages to the Ni-complex: the stability of the catalyst was extended dramatically, both in the absence and in the presence of $O_2$. While previous strategies for

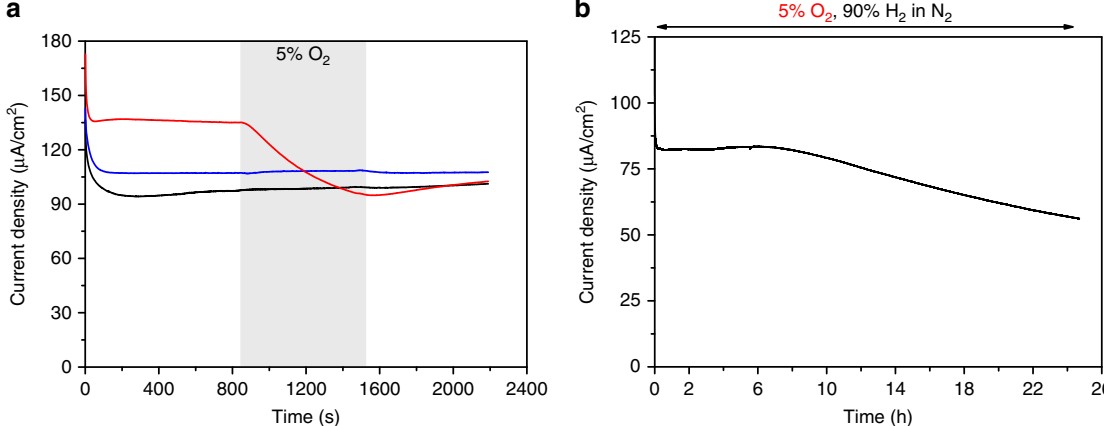

**Fig. 4** Catalytic $H_2$ oxidation by CyGly/polymer films in the presence of $O_2$. **a** Chronoamperometric experiments of glassy carbon electrodes (GCEs) modified with CyGly/polymer films of different thicknesses obtained by drop-casting 20 μL (black line), 10 μL (blue line) and 2.5 μL (red line) of the CyGly/polymer solution in acetonitrile onto the GCEs. Measurements conducted under 90% $H_2$/10% $N_2$ with addition of 5% $O_2$ (90% $H_2$/5% $N_2$) over 10 min (shaded area). Conditions: + 541 mV vs. SHE, pH = 3, 0.1 MES/0.1 M HEPES + HClO$_4$, 25 °C, 500 rpm. **b** Chronoamperometric experiment with a CyGly/polymer film under 5% $O_2$/90% $H_2$/5% $N_2$ gas mixture over 24 h showing the high stability of the $H_2$ oxidation process in this film even under aerobic conditions. Conditions: + 541 mV vs. SHE, pH = 3, 0.1 M MES/0.1 M HEPES + HClO$_4$, 25 °C, 2000 rpm

protection of enzymes against oxidative damage were based on redox polymers with intrinsic $O_2$ reducing capabilities[24,27–29], here the polymer is redox silent and does not contribute directly to protection via a redox process. Instead, the polymer serves as a supporting matrix that imposes local redox states and confers local reactivity to the embedded catalysts. For sufficiently thick films, the polymer electrically isolates the catalyst in the outer layer (protection layer) from the electrode. As a result, when exposed to $H_2$, this portion of the catalyst remains in the $Ni^0$ state and is capable of reducing incoming $O_2$. This allows the catalyst to effectively eliminate $O_2$ in the outer layers of the polymer, preventing it from reaching the electrocatalytic $H_2$ oxidation layer. Moreover, it is possible to achieve high catalyst concentrations inside the polymer film to favor fast interaction of $O_2$ with two reduced Ni-complexes to ensure the complete reduction of $O_2$ to water. It is the combination of these effects that makes the $O_2$ protection mechanism possible.

This possibility to activate the catalyst for self-protection without the need for auxiliary functionalities greatly simplifies the protection concept. Importantly, the ability to maintain the metal complex in a highly reduced protonated state, which is essential for protection, scales with the activity for $H_2$ oxidation, implying that future highly active catalysts, even if potentially highly oxygen sensitive, may still find technological application by exploiting this protection concept.

## Methods

**Electrochemical experiments**. All electrochemical experiments, electrode modifications and handling of the CyGly were carried out inside a glovebox (MBRAUN) filled with nitrogen. A set of mass flow-controllers (Brooks Instruments) were used to control the gas composition flushed through the electrochemical cell. The total flow in all experiments was 1000 mL min$^{-1}$, unless stated otherwise. An oxygen filter (Air Liquide) before the electrochemical cell avoids any undesired $O_2$ contamination. The potential was controlled by a VersaSTAT 4-400 potentiostat. A standard three electrode water jacketed electrochemical cell was used for the measurements with a Pt wire as counter electrode and a saturated calomel electrode located in a side arm as reference electrode. All potentials were converted to the standard hydrogen electrode (SHE) by adding + 241 mV.

**NMR experiments**. A DPX200 NMR spectrometer from Bruker with a proton resonance frequency of 200.13 MHz was used for polymer characterization and a Bruker Avance500 (202.4 MHz $^{31}$P resonance frequency) for $^{31}$P NMR measurements. Solution state $^2$H NMR spectra were recorded on an Agilent VNMR spectrometer (500 MHz $^1$H resonance frequency). Direct detect dual-band or OneNMR probes were used. Typical $^2$H 90° pulses were ~10 μs. The polymer was characterized in deuterated acetone-d6. The residual solvent peak was used as the internal standard.

**Ni complex and polymer synthesis**. The synthesis of the CyGly complex was described earlier[10]. All chemicals and solvents for polymer synthesis were purchased from Sigma-Aldrich, Alfa Aesar, Acros-Organics or J.T. Baker. The free radical initiator 2,2'-azobis(2-methylpropionitrile) (AIBN) was recrystallized from hot toluene prior to use. The co-monomers glycidyl methacrylate (GMA), butyl acrylate (BA) and poly(ethylene glycol methacrylate) (dissolved in isopropyl alcohol, 0.05 g mL$^{-1}$) were passed through a short column filled with inhibitor remover prior to polymerization. The initiator and the co-monomers were stored at −20 °C or +4 °C.

**Size exclusion chromatography (SEC)**. SEC measurements were conducted against polystyrene standards in THF at 30 °C. Data were analyzed with the PSS WinGPC Unity software. Sample concentration was 15 mg mL$^{-1}$.

The redox silent polymer matrix poly(glycidyl methacrylate-co-butyl acrylate-co-poly(ethylene glycol methacrylate)) (P(GMA-BA-PEGMA)) was synthesized following protocols described previously (see Supplementary Fig. 13)[30]. The co-monomers glycidyl methacrylate (GMA, 0.68 mL, 0.710 g, 5 mmol, 49.5 mol%), butyl acrylate (BA, 0.61 mL, 0.545 g, 4.3 mmol) and poly(ethylene glycol methacrylate) (PEGMA, $M_n$ = 500 g mol$^{-1}$, 8 mL of a 0.05 g mL$^{-1}$ isopropyl alcohol solution, 0.4 g, 0.8 mmol, 8 mol%) were mixed in a Schlenk tube under argon atmosphere. Then, the free radical initiator AIBN (15 mg, 91 μmol) was added and the mixture was deaerated by bubbling argon through the solution. The reaction mixture was heated to 80 °C and stirred for ≈35 min at this temperature. The turbid solution was quenched with 40 mL of water. The colorless precipitate was separated by means of a centrifuge (4000 rpm, 20 min). The supernatant was decanted off and the residue was successively washed with water (40 mL), MeOH/water (40 mL, 1:1 vol%) and finally with diethyl ether (120 min) with centrifugation and separation after each step. The highly viscous colorless product was dried under reduced pressure. The dry foamy solid was dissolved in acetonitrile (14.5 mL) to obtain a polymer solution with a concentration of 0.1 g mL$^{-1}$ (yield: 1.45 g, 88%). $^1$H-NMR (200.13 MHz, acetone-d$_6$) δ/ppm: ≈1 (broad, -CH$_3$ and -CH$_2$- of backbone); 1.41 (m, -CH$_2$-, BA); 1.62 (m, -CH$_2$-, BA); 2.69, 2.84 and 3.26 (all s, epoxide, GMA), 3.60 (s, -CH$_2$-, PEGMA), 3.82 and 4.37 (all broad, -CH$_2$-O moiety, GMA), 4.02 (broad, -CH$_2$-O moiety, BA) composition determined via the integral ratio extracted from the $^1$H-NMR spectrum: GMA = 60 mol%, BA = 36 mol%, PEGMA = 4 mol% (Supplementary Fig. 14); SEC (vs. poly(styrene) standard, THF): $M_w$ = 39 kDa, PDI = 2.8.

**Film formation**. Glassy carbon electrodes (GC, Pine Research Instrumentation, 5 mm diameter) were drop-cast with a mixture of 1 mg of CyGly and polymer solution (20 μL, 20 mg mL$^{-1}$) in acetonitrile) and left to dry for 1 h in the glove box. Immobilization of the catalyst on a monolayer was performed as described elsewhere[15]. The pH 3 electrolyte used in all experiments was a 0.1 M MES/0.1 M HEPES mixture where the pH was adjusted with the required amount of HClO$_4$.

**Data availability**. All data are available from the authors upon reasonable request.

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

## Acknowledgements

We like to thank Birgit Nöring and Karolina Lewandowska for technical assistance in electrocatalytic measurements and Prof. S. Ludwigs and Dr. K. Dirnberger (both University of Stuttgart, IPOC) for size exclusion chromatography experiments. This work was funded by the Max Planck Society, the US Department of Energy (DOE, Office of Science Early Career Research Program), the Cluster of Excellence RESOLV (EXC1069) from the Deutsche Forschungsgemeinschaft (DFG) and the Deutsch-Israelische Projektkooperation (DIP) in the framework of the project "Nanoengineered optoelectronics with biomaterials and bioinspired assemblies" funded by the Deutsche Forschungsgemeinschaft (245916028). N. Plumeré acknowledges funding by the European Research Council (ERC Starting Grant 715900). Pacific Northwest National Laboratory is operated by Battelle for the US DOE.

## Author contributions

A.A.O. performed all electrocatalytic experiments with the polymer and $^{31}$P NMR spectroscopy, P.R.-M. the electrocatalytic measurements with the catalyst-monolayer. A.R. designed, synthesized and characterized the polymer. N.P.B. synthesized the CyGly catalyst and performed $^2$H NMR spectroscopy to determine the O$_2$ reduction TOF. All authors contributed to the manuscript writing, experiment design and discussion of the results.

## Additional information

**Competing interests:** The authors declare no competing financial interests.

