## [Peer Review File · Nature Communications]

Reviewers' comments:

Reviewer #1 (Remarks to the Author):

The major focus of this submission is the development of a first-row transition metal electrocatalyst for proton reduction. The Dubois diphosphine,diamine ligand/Nickel complex, (P₂N₂)₂Ni(II/0), has become famous for its rapid rates of proton reduction to dihydrogen. Furthermore, it can also be configured for the oxidation of H₂ as required in fuel cells. Unfortunately for such applications, as a homogeneous electrocatalyst it is rather fragile and notably O₂ sensitive. Hence the goal of this work is to imbed a glycine (N-CH₂C(O)H)modified version of the Dubois catalyst into a polymer film, adhered to a glassy carbon electrode, leading to thermal stability and the potential for repair of O₂ degradation or protection against O₂ damage. The comparison of two versions of this polymer film as well as the complex that is simply adhered to the electrode provides evidence for the existence of two regimes of the film on the electrode. This conclusion is convincingly backed up by the data and should inspire others who pursue electrocatalysis to apply this approach/procedure. Overall the approach is creative and the result of stability (18 days working electrode and increased tolerance to O₂) gives hope that further development would be successful.

There are some questions that this reviewer would like addressed:

Is there not an alternate way to measure the mobility of H₂ vs. O₂ through the polymer in order to give support to the claim that it is not physical blocking of O₂ that gives rise to the stabilization effect?

With respect to Figure 1, Why does the electrode i have nearly the same activity profile as ii? The polymer should have had some stabilizing effect or O₂ protection effect even if not configured the same as iii.

The authors should comment on the initial drop in current for both ii and iii.

Scheme 1 needs a minor modification in that R = glycyI should have only one carbon. perhaps the second line to a presumed carbon should be dashed? or squiggly line?

Publication is recommended.

Reviewer #2 (Remarks to the Author):

What are the major claims of the paper?

The work describes the fabrication and analysis of an H₂ oxidation electrode for catalyzing a fuel cell reaction under ambient O₂ conditions. The electrode is composed of a carbon anode with a Ni-catalyst embedded in a polymer film. The Ni-catalyst is a previously developed H₂ oxidation catalyst derived from the DuBois Ni-N₂P₂ framework. The polymer serves two functions; (i) two-dimensional scaffold to support the catalyst loading, and (ii) a partial diffusion barrier to O₂. The aim of the approach was to balance the catalyst loading with reactivity to H₂ and O₂ in order to allow H₂ oxidation at the anode to drive current, while also enabling the H₂-dependent O₂ reduction to water that required high catalyst density to support formation of binuclear complexes (or intermolecular exchange of the 2-electron reduction product, H₂O₂, between two catalysts?) to perform 4-electron reduction of O₂.

The resulting current densities at $\mu\text{A}/\text{cm}^2$, which are low, but the fact this fuel cell device is composed entirely of Ni (as opposed to Pt or other precious metal based catalysts) is novel. The chemistry of the device further promotes balancing of H generated from H₂ oxidation that would otherwise reduce the pH of the polymer/catalysts film and possible cause reduction in current densities from kinetic effects of high [H⁺] or pH effects on Ni-catalyst stability.

Overall the work is a step forward in fuel cell design using established Ni-based H₂ oxidation catalysts, and thus sets forward a path for further work by this group and others on engineering of these catalysts to further optimize function specific to this device design.

I recommend publication.

Are they novel and will they be of interest to others in the community and the wider field?

Yes, the work is innovative and novel. Fuel cell device development is of broad interest to fundamental and applied sciences, and engineering.

On a more subjective note, do you feel that the paper will influence thinking in the field?

Yes, this is a clever, though inventive approach and offers a intriguing design principle for advancing fuel cell device design, especially if higher current densities can be achieved in future iterations. This work establishes a clear path forward and should be highly cited.

Reviewer #3 (Remarks to the Author):

This manuscript describes an interesting electrocatalytic system for H₂ oxidation using an O₂-sensitive Ni complex. The authors dispersed the Ni catalyst in a redox-silent hydrophobic polymer, and modified the catalyst/polymer mixture on a glassy carbon electrode. The activity of the catalyst-modified electrode for electrocatalytic H₂ oxidation was evaluated in the absence/presence of O₂. As a result, they found that stability for electrocatalytic H₂ oxidation and O₂ tolerance of the catalyst can be enhanced by dispersing into the polymer matrix, and concluded that Ni complexes close to the electrode surface serve as catalysts for H₂ oxidation and those at the outer film boundary catalyze O₂ reduction into H₂O. This conclusion is highly interesting. However, in the present manuscript, the conclusion is not fully substantiated by their experimental results. In particular, although the authors proposed that the Ni complex can catalyze O₂ reduction into H₂O in the polymer matrix, the activity of the Ni complex for the reaction was evaluated only in solution and there is no experimental evidence of the reaction in the polymer matrix. In addition, the turnover frequency of the Ni catalyst for O₂ reduction is very slow (TOF = ca. 20 per hour). I cannot believe that such a slow reaction can sufficiently decrease O₂ gas in the polymer matrix. It can also be considered that O₂ gas in the polymer matrix is just consumed during the degradation of the O₂-sensitive Ni catalyst. On balance, I cannot recommend this manuscript for publication in Nature Communications at this stage. I encourage the authors to resubmit their revised manuscript after obtaining extensive experimental results to substantiate their claim.

Responses to Reviewers' Comments:

Reviewer #1

The major focus of this submission is the development of a first-row transition metal electrocatalyst for proton reduction. The Dubois diphosphine,diamine ligand/Nickel complex, (P2N2)2Ni(II/0), has become famous for its rapid rates of proton reduction to dihydrogen. Furthermore, it can also be configured for the oxidation of H2 as required in fuel cells. Unfortunately for such applications, as a homogeneous electrocatalyst it is rather fragile and notably O2 sensitive. Hence the goal of this work is to imbed a glycine (N-CH2C(O)2H)modified version of the Dubois catalyst into a polymer film, adhered to a glassy carbon electrode, leading to thermal stability and the potential for repair of O2 degradation or protection against O2 damage. The comparison of two versions of this polymer film as well as the complex that is simply adhered to the electrode provides evidence for the existence of two regimes of the film on the electrode. This conclusion is convincingly backed up by the data and should inspire others who pursue electrocatalysis to apply this approach/procedure. Overall the approach is creative and the result of stability (18 days working electrode and increased tolerance to O2) gives hope that further development would be successful.

We are thankful to reviewer 1 for his positive general comments, we will address point by point his specific remarks.

There are some questions that this reviewer would like addressed:

Is there not an alternate way to measure the mobility of H₂ vs. O₂ through the polymer in order to give support to the claim that it is not physical blocking of O₂ that gives rise to the stabilization effect?

As the referee points out, we were interested to test the mobility of O₂ vs H₂ in the polymer. It is clear that the polymer hinders gas diffusion to the electrode surface as it is shown in Figure S4. Nevertheless, as we show on Figure 1B, when a monolayer of catalyst is coated with polymer only, the catalytic activity is rapidly lost, indicating that the physical blockage does not provide enough protection. In addition to this, in Figure S6 (SI) we show that the catalyst loading plays a

crucial factor in the O₂-protection process where the films with low catalyst loading or without any catalyst suffer from rapid degradation. Hence, the presence of a distinct amount of catalyst in the entire polymer film (especially in the outer layer of the polymer film) is a prerequisite for effective protection, which cannot be attributed to O₂ physical blockage only.

With respect to Figure 1, why does the electrode i have nearly the same activity profile as ii? The polymer should have had some stabilizing effect or O₂ protection effect even if not configured the same as iii.

We were as surprised as the reviewer about this result. In the end, we are analyzing the activity of a monolayer of catalyst, with a coverage on the order of several picomol/cm². The activity profile of ii highlights the extreme O₂ sensitivity of the catalyst under H₂ oxidation conditions. As mentioned above, this result highlights the relevance of the catalyst role in the protection layer to change the catalytic profile to what is shown in (iii). To answer a question addressed by Reviewer #3 we decided to modify this experiment to highlight the importance of having a catalytically active Ni complex. Now trace (ii) corresponds to a monolayer of catalyst in the DET regime coated with polymer loaded with the same ligand that coordinates the Ni complex, but without the metal center. In that case the O₂ damages the complex at a slightly slower rate than for trace (i), but much faster than for (iii). This confirms that O₂ oxidation of the ligands just plays a marginal role in the catalyst protection mechanism.

We have added-modified the following paragraph to the main text:

However, when a monolayer of the catalyst on the electrode surface was further coated with a mixture of polymer and ligand (of similar thickness, with two equivalents of CyGly ligand but without Ni), the catalyst exhibits similar sensitivity to oxygen as the monolayer without a polymer over layer (Figure 1B, red trace). This excludes on one hand the possibility that a physical barrier to oxygen is the sole explanation for the decreased oxygen sensitivity in catalyst-polymer films. On the other hand, since the CyGly ligand is also O₂ sensitive, this experiment further demonstrates that the simple stoichiometric reaction of O₂ with the catalyst during its oxidative degradation does not significantly participate in the protection mechanism. An active Ni-complex dispersed in the polymer film is required for effective protection.

The authors should comment on the initial drop in current for both ii and iii.

The initial current in (i) and (ii) in Figure 1 is a capacitive charging current generated due to the presence of the polymer film that changes the properties of the solution-electrode interface. As a potential is applied, the charges on both sides come to a balance and the capacitive charging current will drop. In addition to this, inside the polymer film the slow diffusion of the components delay current stabilization. We added a sentence in the main text to clarify this point:

“ The initial current drop corresponds to slow equilibration processes inside the polymer (establishment of H₂ and Ni(II) gradients). ”

Scheme 1 needs a minor modification in that R = glycyl should have only one carbon. perhaps the second line to a presumed carbon should be dashed? or squiggly line?

We thank the reviewer for the suggestion, the structure is now modified accordingly in the revised version.

Publication is recommended.

Reviewer #2:

What are the major claims of the paper?

The work describes the fabrication and analysis of an H₂ oxidation electrode for catalyzing a fuel cell reaction under ambient O₂ conditions. The electrode is composed of a carbon anode with a Ni-catalyst embedded in a polymer film. The Ni-catalyst is a previously developed H₂ oxidation catalyst derived from the DuBois Ni-N₂P₂ framework. The polymer serves two functions; (i) two-dimensional scaffold to support the catalyst loading, and (ii) a partial diffusion barrier to O₂. The aim of the approach was to balance the catalyst loading with reactivity to H₂ and O₂ in order to allow H₂ oxidation at the anode to drive current, while also enabling the H₂-dependent O₂ reduction to water that required high catalyst density to support formation of binuclear complexes (or intermolecular exchange of the 2-electron reduction product, H₂O₂, between two catalysts?) to perform 4-electron reduction of O₂.

The resulting current densities at $\mu\text{A}/\text{cm}^2$, which are low, but the fact this fuel cell device is composed entirely of Ni (as opposed to Pt or other precious metal based catalysts) is novel. The chemistry of the device further promotes balancing of H generated from H₂ oxidation that would otherwise reduce the pH of the polymer/catalysts film and possible cause reduction in current densities from kinetic effects of high [H⁺] or pH effects on Ni-catalyst stability.

Overall the work is a step forward in fuel cell design using established Ni-based H₂ oxidation catalysts, and thus sets forward a path for further work by this group and others on engineering of these catalysts to further optimize function specific to this device design.

I recommend publication.

Are they novel and will they be of interest to others in the community and the wider field?

Yes, the work is innovative and novel. Fuel cell device development is of broad interest to fundamental and applied sciences, and engineering.

On a more subjective note, do you feel that the paper will influence thinking in the field?

Yes, this is a clever, though inventive approach and offers a intriguing design principle for advancing fuel cell device design, especially if higher current densities can be achieved in future iterations. This work establishes a clear path forward and should be highly cited.

We are very thankful to the reviewer for his kind comments on the presented research.

Reviewer #3:

This manuscript describes an interesting electrocatalytic system for H₂ oxidation using an O₂-sensitive Ni complex. The authors dispersed the Ni catalyst in a redox-silent hydrophobic polymer, and modified the catalyst/polymer mixture on a glassy carbon electrode. The activity of the catalyst-modified electrode for electrocatalytic H₂ oxidation was evaluated in the absence/presence of O₂. As a result, they found that stability for electrocatalytic H₂ oxidation and O₂ tolerance of the catalyst can be enhanced by dispersing into the polymer matrix, and concluded that Ni complexes close to the electrode surface serve as catalysts for H₂ oxidation and those at the outer film boundary catalyze O₂ reduction into H₂O. This conclusion is highly interesting. However, in the present manuscript, the conclusion is not fully substantiated by their experimental results.

In particular, although the authors proposed that the Ni complex can catalyze O₂ reduction into H₂O in the polymer matrix, the activity of the Ni complex for the reaction was evaluated only in solution and there is no experimental evidence of the reaction in the polymer matrix.

In addition, the turnover frequency of the Ni catalyst for O₂ reduction is very slow (TOF = ca. 20 per hour). I cannot believe that such a slow reaction can sufficiently decrease O₂ gas in the polymer matrix. It can also be considered that O₂ gas in the polymer matrix is just consumed during the degradation of the O₂-sensitive Ni catalyst.

Concentration in the polymer matrix is much higher than in solution. Thus, the reduction, for which two molecules of the reduced Ni-complex are required, seems to be much faster. This is supported by the results obtained for polymer/catalyst electrodes modified with different catalyst loadings: decreasing the Ni-catalyst concentration leads to less efficient protection (see Figure S6). Hence, the polymer not only ensures the immobilization of the catalyst but also ensures a high concentration of the catalyst in the hydrophobic protection matrix which is substantially higher than the concentration of the catalyst in water.

On balance, I cannot recommend this manuscript for publication in Nature Communications at this stage. I encourage the authors to resubmit their revised manuscript after obtaining extensive experimental results to substantiate their claim.

In order to answer these comments we have designed two additional experiments:

-We have modified the experiments in Figure 1. Now trace (ii) corresponds to a monolayer of catalyst in the DET regime coated with polymer loaded with the same ligand that coordinates the Ni complex, but without the metal center. To account for the same number of reactive species we added two equivalents of ligand with respect to the amount of Ni complex used in (iii). In that case the O₂ damages the complex at a slightly slower rate than for trace (i), but much faster than for (iii). The ligands include the phosphines, which have been proposed to be the most sensitive position to the attack by O₂ (Yang, J.Y. *et al.* Dalton Trans. 2010; Wakerley *et al.* Chem. Comm. 2014). This confirms that O₂ oxidation of the ligands, or catalyst degradation as the referee suggests, does not offer effective protection and the presence of a catalytically active species in the polymer film is required.

-We tested the stability of the catalyst inside the polymer film when exposed to O₂ in the presence and in the absence of H₂. The proposed protection mechanism requires the catalyst to be reduced by H₂ in order to allow the catalytic consumption of O₂ by the Ni complex. Indeed, when exposed to O₂ in the absence of H₂ the catalyst loses its catalytic activity. In Supplementary Figure S8 we show experimental evidence that supports the catalytic nature of the protection mechanism. We added the following paragraph in the main text:

To further investigate the catalytic nature of the O₂ reduction reaction by the catalyst, we tested the stability of the catalytic current when the film was exposed to O₂ in the absence of H₂. This gas is required to regenerate the reduced Ni-complex in the protection layer once it is oxidized by an O₂ molecule. Supplementary Figure S8 shows that when the catalyst-polymer film is exposed to a 5% O₂ in the absence of H₂ for 5 hours, 35% of the catalytic activity is lost, while when H₂ is present, 100% of the initial current is maintained for the same O₂ exposure. Interestingly, while measuring the control experiment under N₂ (Supplementary Figure S8, blue trace), without O₂ to corroborate that N₂ is not damaging the complex, we noticed that the oxidation current did not reach 0 A. The most likely explanation for this observation is that as we start the experiment under H₂ to record the initial activity of the film, the catalyst in the polymer layer is completely reduced by H₂. Since diffusion inside the polymer is extremely slow, its oxidation by the electrode surface is also very slow. Integration of the charge passed gives a value of 31 μC, which would correspond to 0.2 μmol of Ni-complex, 20% of the catalyst present in the film. This explains why only 35% of the catalytic activity is lost under a N₂/O₂ mixture. Under such conditions, the Ni-complex in the polymer film is still reduced, and therefore protecting the complex in the inner layer, but in the absence of H₂ to regenerate the reduced complex, the inactive front advances more rapidly.

And the following figure in the SI:

Figure S8. Chronoamperometry experiments of three CyGly/polymer films deposited on GCE. The initial gas mixture is 90 % H₂ in N₂ for all three experiments to evaluate the catalytic current before O₂ exposure. The currents were normalized to this value. The black trace corresponds to an electrode that was exposed to a 5% O₂, 90% H₂ and 5% N₂ mixture for 5 h. For the red and blue traces, after 30 min. the gas was switched to 100% N₂ and for the red trace a 5% O₂ was added to the gas mixture for 5 h. On the final step, the gas flow was switched back to the initial 90% H₂ in N₂ composition. Conditions: +541 mV vs. SHE, 25°C, pH = 3, 1000 rpm.

Reviewers' Comments:

Reviewer #1 (Remarks to the Author):

In my opinion the authors have conscientiously addressed all reviewer concerns, including my own. The manuscript should be published in its revised form.

Reviewer #3 (Remarks to the Author):

I have read the revised manuscript and the point-by-point response to the reviewers' comments. Now I am satisfied with the revisions made by the authors, and I would recommend publication of the revised manuscript in Nature Communications.